# The 'ForensOMICS' approach for postmortem interval estimation from human bone by integrating metabolomics, lipidomics, and proteomics

Andrea Bonicelli[1]*[†], Hayley L Mickleburgh[2,3], Alberto Chighine[4], Emanuela Locci[4], Daniel J Wescott[3], Noemi Procopio[1,3]*[†]

[1]The Forensic Science Unit, Faculty of Health and Life Sciences, Northumbria University, Newcastle upon Tyne, United Kingdom; [2]Amsterdam Centre for Ancient Studies and Archaeology (ACASA) – Department of Archaeology, Faculty of Humanities, University of Amsterdam, Amsterdam, Netherlands; [3]Forensic Anthropology Center, Texas State University, San Marcos, United States; [4]Department of Medical Science and Public Health, Section of Legal Medicine, University of Cagliari, Monserrato, Italy

*For correspondence: ABonicelli@uclan.ac.uk (AB); NProcopio@uclan.ac.uk (NP)

Present address: [†]School of Natural Sciences, University of Central Lancashire, Preston, United Kingdom

Competing interest: The authors declare that no competing interests exist.

**Abstract** The combined use of multiple omics allows to study complex interrelated biological processes in their entirety. We applied a combination of metabolomics, lipidomics and proteomics to human bones to investigate their combined potential to estimate time elapsed since death (i.e., the postmortem interval [PMI]). This 'ForensOMICS' approach has the potential to improve accuracy and precision of PMI estimation of skeletonized human remains, thereby helping forensic investigators to establish the timeline of events surrounding death. Anterior midshaft tibial bone was collected from four female body donors before their placement at the Forensic Anthropology Research Facility owned by the Forensic Anthropological Center at Texas State (FACTS). Bone samples were again collected at selected PMIs (219-790-834-872days). Liquid chromatography mass spectrometry (LC-MS) was used to obtain untargeted metabolomic, lipidomic, and proteomic profiles from the pre- and post-placement bone samples. The three omics blocks were investigated independently by univariate and multivariate analyses, followed by Data Integration Analysis for Biomarker discovery using Latent variable approaches for Omics studies (DIABLO), to identify the reduced number of markers describing postmortem changes and discriminating the individuals based on their PMI. The resulting model showed that pre-placement metabolome, lipidome and proteome profiles were clearly distinguishable from post-placement ones. Metabolites in the pre-placement samples suggested an extinction of the energetic metabolism and a switch towards another source of fuelling (e.g., structural proteins). We were able to identify certain biomolecules with an excellent potential for PMI estimation, predominantly the biomolecules from the metabolomics block. Our findings suggest that, by targeting a combination of compounds with different postmortem stability, in the future we could be able to estimate both short PMIs, by using metabolites and lipids, and longer PMIs, by using proteins.

## Editor's evaluation

This well-presented and sophisticated study provides significant proof-of-concept for the application of the ForensOMICS approach as a new pathway for forensic taphonomy with great promise to advance future research. The solid foundation of the research combining metabolomics,

proteomics, and lipidomics is considered very exciting, strong, and expands the boundaries of forensics research.

## Introduction

The modifications that occur to the human body after death are complex and known to be affected by a variety of intrinsic and extrinsic factors. The rate of decomposition can vary significantly depending on the environment and even the manner of death. Nonetheless, the process of decomposition has been demonstrated to be predictable, providing opportunities to estimate the time elapsed since death (also known as postmortem interval [PMI]) based on gross morphological and/or microscopic changes to the body. Precise and accurate estimation of the PMI is crucial to help establishing the timeline of events surrounding death, can support medicolegal investigators with the identification of the deceased and can corroborate or negate other forensic evidence.

In the first hours after death, the body undergoes several postmortem changes, including progressive cooling (*algor mortis*), increased rigidity associated with muscle stiffness (*rigor mortis*), and pink-purplish discolouration, in light skinned individuals, caused by the lack of blood circulation and the settling of blood in the lowest areas (*livor mortis*) (*Clark et al., 1997*; *Henssge and Madea, 2007*; *Madea, 2016*). After these stages, as the time since death increases, the breaking down and liquefaction of the organs and other soft tissues will occur: a process referred to as putrefaction. The lack of oxygenated circulation induces cellular hypoxia, leading to swelling of the cells, and subsequent rupture of cell membranes and release of digestive enzymes. This triggers the autolytic digestion of soft tissues (*Lee Goff, 2009*). The body becomes fully anaerobic, allowing anoxic (endogenous) bacteria originated from the gut to proliferate and transmigrate throughout the entire body (*Javan et al., 2016*; *Jans et al., 2004*). The activity of endogenous bacteria results in the accumulation of gases which causes bloating of the soft tissues, starting from the abdomen but also taking place in the face during early decomposition stages, and progressing towards the rest of the body. Colonization of the body by insects and exogenous bacteria, mostly aerobic microorganisms, contributes further to the macroscopic changes and to the reduction of the overall mass of soft tissues (*Hyde et al., 2013*; *Carter et al., 2017*). Besides these, other extrinsic factors including abiotic environmental conditions (e.g., humidity, temperature, sun exposition, aeration, burial context) and biotic factors, such as the presence and type of microorganisms, insects, and scavengers (*Cockle and Bell, 2015*; *Procopio et al., 2017b*), will affect the rate of decomposition of the soft tissues. Intrinsic factors known to affect the rate of decomposition include, among others, body mass index and both antemortem and perimortem pathological conditions (*Mickleburgh et al., 2021*). Completion of the putrefactive stage and the activity of insects consuming the remaining soft tissues will leave the remains completely, or almost completely, skeletonized, and dry.

The complex nature and interplay of intrinsic and extrinsic variables involved in the process of decomposition makes the development of accurate and precise models for PMI estimation extremely challenging. Traditional methods of PMI estimation include calculating PMI using the body temperature and ambient temperature (which relies on the predictability of *algor mortis*, and works for short PMIs only), or the visual assessment of gross morphological changes to the body to estimate a relatively wide PMI range. Since the rate of gross morphological changes is variable, methods that rely on the visual scoring of decomposition stages suffer from issues of poor accuracy and precision. An additional problem of such methods is the effect of interobserver variations on the scoring of decomposition stages. For all commonly used PMI estimation methods, the accuracy and precision decreases considerably as decomposition progresses, and is particularly problematic when the remains are partially or completely skeletonized (*Henssge and Madea, 2007*; *Madea, 2016*).

In recent years, the number of studies exploring the use of biomolecular methods for PMI estimation has risen sharply, due to their potential for providing more accurate and precise estimation methods based on the rates of decay of different molecules and compounds (*Procopio et al., 2018b*; *Prieto-Bonete et al., 2019*; *Pesko et al., 2020*; *Locci et al., 2019*; *Zelentsova et al., 2020*). Better understanding of biomolecular decomposition of bone will provide opportunities to develop biomolecular methods for the estimation of longer PMIs (i.e., timeframes in which soft tissues are unlikely to be preserved). Moreover, through the combined analysis of multiple different panels of omics, greater precision and accuracy of PMI estimation can potentially be achieved.

Biomolecular decomposition is caused by both enzymatic and microbial breakdown of large molecules, resulting in the breakage of proteins into amino acids (AA), of carbohydrates into more simple monosaccharides, and of lipids into simpler fatty acids chains (*Dent et al., 2004*; *Nolan et al., 2020*). In carbohydrate decomposition, the complex polysaccharides are normally broken down via microbial activity into smaller units of monosaccharides. This breakdown can be fully achieved by oxidation, with the production of carbon dioxide and water, or can be partially achieved with the production of organic acids and alcohols. Alternatively, the monosaccharides can be degraded by fungal activity into glucuronic, citric, and oxalic acids, or by bacteria into lactic, butyric, and acetic acids (*Dent et al., 2004*; *Stuart, 2013*). During the decay of lipids, free saturated and unsaturated fatty acids are released due to the hydrolysis mediated by the action of intrinsic lipases released after death. These can then be converted into hydroxyl fatty acids (the main constituent of adipocere) by the action of specific bacterial enzymes in humid environments, or can associate with potassium and sodium ions, resulting in the formation of salts (*Stuart, 2013*). Protein degradation is primarily an enzyme-driven process, led by the action of proteases, which occurs at different rates for different proteins and tissues. Proteolytic enzymes induce the hydrolytic breakdown of proteins and the production respectively of proteoses, peptones, polypeptides, and finally AA, which can be further modified via deamination (production of ammonia), decarboxylation (production of cadaverine, putrescine, tyramine, tryptamine, indole, skatole, and carbon dioxide) and desulfhydralation (production of hydrogen sulphide, pyruvic acid, and thiols) (*Dent et al., 2004*; *Stuart, 2013*). Time-dependent non-enzymatic processes can also affect protein degradation and modifications (i.e.,deamidations).

The analysis of low molecular weight compounds and decomposition by-products is becoming more popular in forensic science, particularly for the purpose of estimating PMI (*Locci et al., 2020*). Time since death was recently reported as the main variable driving modifications in the metabolome occurring after death (*Chighine et al., 2021*) in many soft tissues and fluids, so the metabolomic approach appears ideal to estimate PMI. However, the potential forensic significance of the postmortem bone metabolome is as yet underexplored (*Alldritt et al., 2019*). Several studies on soft tissues (vitreous and aqueous humour) have examined metabolomics for the purpose of determining short PMIs. Examining longer PMIs based on metabolomics analysis of humour has not been possible due to evaporation and leakage through the corneal surface as time since death progresses (*Locci et al., 2019*). *Girela et al., 2008* reported a significant positive correlation between PMI and taurine, glutamate, and aspartate levels found in vitreous humour. These results were partially confirmed by *Zelentsova et al., 2020*, who found a correlation between the levels of hypoxanthine, choline, creatine, betaine, glutamate, and glycine and PMI. Another approach employing [1]H-NMR on aqueous humour from pig heads reported taurine, choline, and succinate as major metabolites involved in the postmortem modification (*Locci et al., 2019*). The same study also showed an orthogonally constrained PLS2 model showing prediction error of 59 min for PMI <500 min, 104 min for PMI from 500 to 1000 min, and 118 min for PMI >1000 min. Besides humour, muscle is one of the most frequently targeted tissues in metabolomics studies focused on short PMI estimation. *Pesko et al., 2020* recently evaluated rat and human biceps femoris muscles from the same individuals at different PMIs, demonstrating an increase of the abundance of several metabolites, including most of those derived from the breakdown of proteins, and in particular highlighting how threonine, tyrosine and lysine show the most consistent and predictable variations in relatively short PMIs. An untargeted metabolomics study on muscle tissue also indicated the potential of isolating biomarkers associated with age (*Wilkinson et al., 2020*), suggesting the potential applications of metabolomics for both age-at-death and PMI estimation.

To date, only three studies have used lipidomics assays for PMI estimation. Two of them were conducted on muscle tissue and showed, in general, a negative correlation between most lipid classes and PMI, as well as an increment in free fatty acids (*Langley et al., 2019*; *Wood and Shirley, 2013*). The third study applied lipidomics to trabecular bone samples from calcanei spanning a PMI of approximately 7 years and highlighted the presence of 76 potential *N*-acyl AA that could be employed for PMI estimation, however their correlation with PMI has not yet been fully elucidated (*Dudzik et al., 2017*).

Several studies have tried to quantify the degree of survival of proteins and the accumulation of post-translational modifications (PTMs) of AA in both animal and human models (*Procopio et al., 2017b*; *Mickleburgh et al., 2021*; *Prieto-Bonete et al., 2019*; *Procopio et al., 2021*; *Mizukami*

*et al., 2020*) as well as under different conditions (e.g., in aquatic environments, different types of coffins, buried vs. surface) (*Procopio et al., 2021*; *Mizukami et al., 2020*; *Bonicelli et al., 2022*). The premise of these studies is that the protective action of the hydroxyapatite is expected to enhance the survival of proteins, allowing potential estimation of longer PMIs. Results generally showed that blood/plasma and ubiquitous proteins decrease in their abundance constantly starting from the early decomposition stages, whereas proteins more strongly connected to the mineral matrix such as bone-specific proteins are able to survive for longer PMIs and can be useful indicators for PMI estimation also in skeletonized remains. Similarly, also the accumulation of specific non-enzymatic PTMs, such as deamidations, can be used as a biomarker for the evaluation of the PMI in bones.

While many studies have applied different analytical platforms for proteomics, metabolomics, and lipidomics to several different matrices (*Pesko et al., 2020*; *Locci et al., 2019*; *Zelentsova et al., 2020*; *Girela et al., 2008*; *Li et al., 2018*; *Wu et al., 2018*; *Hirakawa et al., 2009*; *Banaschak et al., 2005*; *Li et al., 2017*), relatively little is known about the biomolecular decomposition of bone tissue. Moreover, while clinical studies have applied multi-omics methods with some frequency, their potential for the development of more precise and accurate biomolecular PMI estimation methods has not been explored. The present study applies, for the first time, a multi-omics approach (i.e., combined proteomics, metabolomics and lipidomics, defined here as the 'ForensOMICS' approach) to pre- and post-decomposition tibial cortical bone samples from four human female body donors, to identify potential multi-omics biomarkers of time since death. The multi-omics approach uses the natural differences in manner and rate of decomposition between the different biomolecules (proteins, metabolites, lipids) to expand the potential range of PMIs and to cross-correlate results between different sets of biomarkers to narrow down PMI ranges based on the degradation of multiple biomolecules. The use a of a single omics technique would not be suitable to investigate a wide range of potential PMIs. Metabolites and lipids are appropriate for short PMIs while protein has been proved to be stable across longer ones. Therefore, the combination of the three classes of biomolecules aims to obtain an ideal coverage across a wider range of PMIs. Additional advantages of the integration of different biomolecule classes might include greater flexibility in their application across different environments and different postmortem treatments, since it could increase the likelihood of retrieving suitable markers for PMI estimation. The present study provides a proof-of-concept for future validation of the multi-omics approach on a larger number of individuals.

## Results

### Single omics profile

The metabolites matrices resulting from the combination of metabolomics ESI+ and ESI- data were combined in a final matrix with a total of 104 identified compounds after the removal of non-endogenous compounds following querying in HMDB. Furthermore, after preliminary inspection via PCA, lipidomics ESI+ results were excluded due to their poor contribution to a potential discriminant model. Each omics block was then evaluated individually via univariate (Kruskal-Wallis and Dunn's pairwise test) and multivariate (partial least square discriminant analysis [PLS-DA]) analysis. The overall the clustered image map (CIM) and individual plot obtained with metabolomics suggested a clear separation between fresh and decomposed samples and the total variance explained by the model in the first two components taken together was 60% (*Figure 1—figure supplement 1*). More interestingly, increasing PMIs were found to cluster progressively further away from the fresh samples. By observing the clustering of the variables in the CIM, it was clear the presence of three major behaviours: (i) reduction in the intensity of compounds between the pre-deposition samples and the skeletonized ones; (ii) higher intensity of compounds for the 219, 790, 843 days PMI groups; (iii) presence of compounds that specifically were more intense in the 872 days PMI. Examples of these behaviours can be observed in *Figure 1—figure supplement 1*. These compounds were found to be significant for Kruskal-Wallis but were only visually selected (*Figure 1—figure supplement 1*) because of their trend with PMI. However, these results were not fully supported by statistical testing, as pairwise analysis mainly showed significant differences between few PMI groups, specifically between baseline vs. more advanced PMIs (*Figure 1—figure supplement 2*). It is interesting to note that D2 appeared to have a specific profile in the pre-deposition state that clearly differed from the other

donors, therefore potentially affecting the overall clustering and partially hiding the effect of PMI. In contrast, D4 after decomposition showed a distinct profile, likely associated with the prolonged PMI.

Lipidomic profiling (*Figure 1—figure supplement 2*) showed that the closer cluster to the pre-deposition individuals is the 872 days group, followed by 219, 790, and 834 days. This could be related to the fact that a large number of lipids, not highly abundant in the fresh portion of the sample, was found to be higher in intensity for early PMIs to then progressively decrease. However, a large block constituted mostly by ceramides, was here shown to be highly present in the skeletonized D4 compared to the remaining individuals, suggesting a relationship with PMI. The same three behaviours extrapolated for metabolite features were identified for lipids (*Figure 1—figure supplement 2*). The model for this block explains 73% of the variance in the first two components.

Finally, proteins showed an inferior discriminatory power in comparison with the other classes of molecules according to individual consensus plot (*Figure 1—figure supplement 3*). The variance explained in the model in the first two components was only 35% and, besides the major separation between pre- and post-decomposition, it was not possible to clearly discriminate the various PMIs (*Figure 1—figure supplement 3*). However, with the exception of D3 (834 days PMI), it is clear that the skeletonized samples cluster away from the fresh ones with increasing PMIs. Few proteins evaluated via univariate statistics, however, showed clear visual and significant negative trends in the overall sample (Kruskal-Wallis), although pairwise comparison could not confirm the statistical significance of the difference across PMIs (Dunn's test, *Supplementary file 1*). These proteins were ASPN_HUMAN, H4_HUMAN, HBB_HUMAN, OSTP_HUMAN, VIME_HUMAN. Moreover, what was clear in *Figure 1—figure supplement 3* is the large variation between replicates that could affect the evaluation of the proteins' behaviour with PMI.

## Omics integration

All the 24 human bone samples were included in the omics integration model (*Figure 1*). We firstly evaluated correlations between the omics block using PLS regression. Results for component 1 showed an R value of 0.94 between metabolomics and lipidomics, 0.96 between metabolomics and proteomics and 0.87 between lipidomics and proteomics. Feature selection using the DIABLO method aimed to identify highly correlated and discriminant variables across the three omics. The arrow plot (*Figure 1A*) showed the overall separation between fresh and skeletonized samples, which was mainly developed along the first component. However, it was possible to note that the individual with the longest PMI (D4, 872 days) also clustered away from the remaining skeletonized samples along the second component (*Figure 1B*). The optimal number of components was set at three by means of threefold cross-validation repeated 100 times (*Figure 1B*). The overall balanced error remained below 0.4 (*Figure 1—figure supplement 4*). After tuning the model by attributing the same weight to all the omics blocks, the ideal panel of markers selected in the first component that retained most of the covariance of the system includes 14 metabolites, 5 lipids and 5 proteins (*Figure 1C*). These loading plots show that a few metabolite markers have a high loading for different PMIs, whereas both lipid and protein markers have high values particularly for the fresh samples. Considering the individual-omics consensus plots in *Figure 1—figure supplement 5*, metabolite and lipid blocks showed a better segregation between the various PMIs in the skeletonized state in comparison with the protein one. There is, however, overlap in all blocks for these intermediate PMIs.

Multi-omics sample variations between bones from fresh and skeletonized cadavers were also supported by the CIM (*Figure 1D*), which showed a clear separation between the two groups. Most of the compounds selected by the model were highly abundant in the fresh samples and less abundant in the skeletonized ones, although the lower panel of metabolites (in *Figure 1D*) showed an opposite trend. In general, it could be observed that the samples with shorter PMIs (up to 834 days) showed a decline for proteins, lipids, and for eight of the metabolites selected for the PMI model as well as an increase in the remaining six metabolites in comparison with their fresh counterparts. Whereas the decline in the abundance of proteins and lipids in comparison with the fresh samples was similar between all the 12 skeletonized samples, the increase or decrease in the abundance of specific metabolites was more exacerbated in the samples with the longest PMI (872 days) in comparison with the others (*Figure 1D*). To conclude, the model was first cross-validated resulting in a mean standard error of the classification error of 9.67. Additionally, after performing permutation test there was still significant difference in the discrimination between the PMIs (p=0.001).

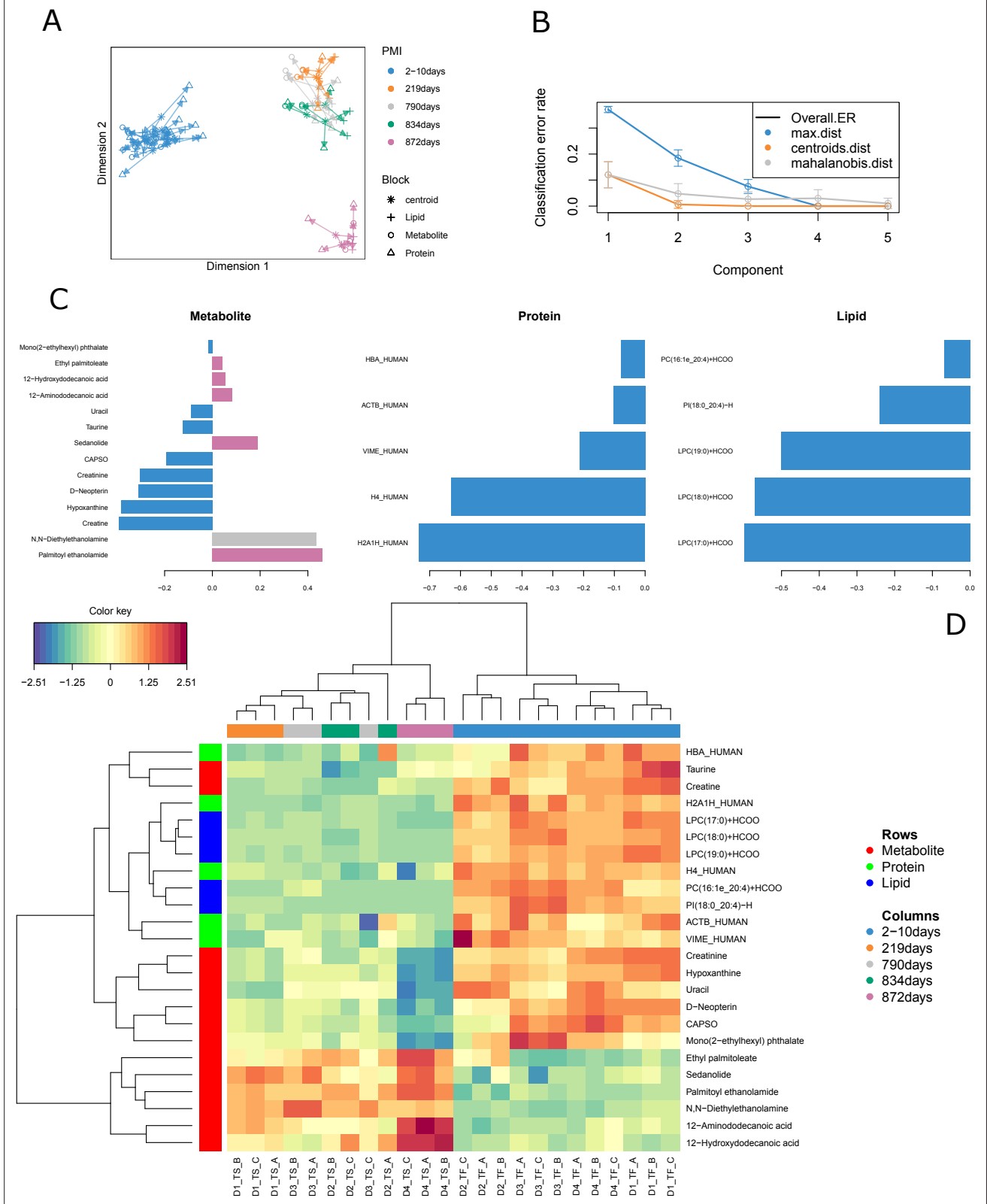

**Figure 1.** Results for the tuned model. (**A**) Arrow plot showing multiblock contexts for the overall model. (**B**) Optimal number of components to explain model variable calculated via cross-validation (error bars provide standard deviation). (**C**) Loading plot showing how each variable contributes to the covariance of each group. (**D**) The clustered image map (CIM) shows the selected compounds in the final model. It is possible to see that most

*Figure 1 continued on next page*

*Figure 1 continued*

compounds decrease in intensity after decomposition except for few metabolites and two lipids that specifically increase in certain postmortem interval (PMI) intervals.

The online version of this article includes the following figure supplement(s) for figure 1:

**Figure supplement 1.** Results for the metabolomics data.

**Figure supplement 2.** Results for the lipidomics data.

**Figure supplement 3.** Results for the proteomics data.

**Figure supplement 4.** Balanced error variations across variable selection steps.

**Figure supplement 5.** Score plots for partial least square discriminant analysis (PLS-DA) results of all the omics blocks considered.

By evaluating individual markers, it was possible to identify compounds that increased or decreased consistently across the PMI (*Figure 2A*). More specifically, palmitoyl ethanolamide, ethyl palmitolate, *N*,*N*-diethylethanolamine, sedanolide, 12-aminododecanoic acid, and acetamide showed the lowest values for the fresh samples and increasing values with prolonged decomposition time. The remaining metabolites decreased consistently with PMI with a considerable drop between the baseline and 219 days. Lipids and proteins selected for the model, instead, were all characterized by a drastic reduction in their intensity in the skeletonized samples in comparison with the fresh ones. Proteins selected here were two histone proteins (histone H2A type 1H [H2A1H] and histone H4 [H4]), haemoglobin subunit alpha (HBA), vimentin (VIME) and actin (ACTB).

High significant correlations (r>0.9) were also identified between compounds belonging to the three distinct omics blocks (*Figure 2B*). Palmitoyl ethanolamide showed negative correlation with all lipids selected but PC(16:1e_20:4)+HCOO and with H2A1H_HUMAN and H4_HUMAN proteins. Creatinine, hypoxanthine, and D-Neopterin were positively correlated with all lipids selected but

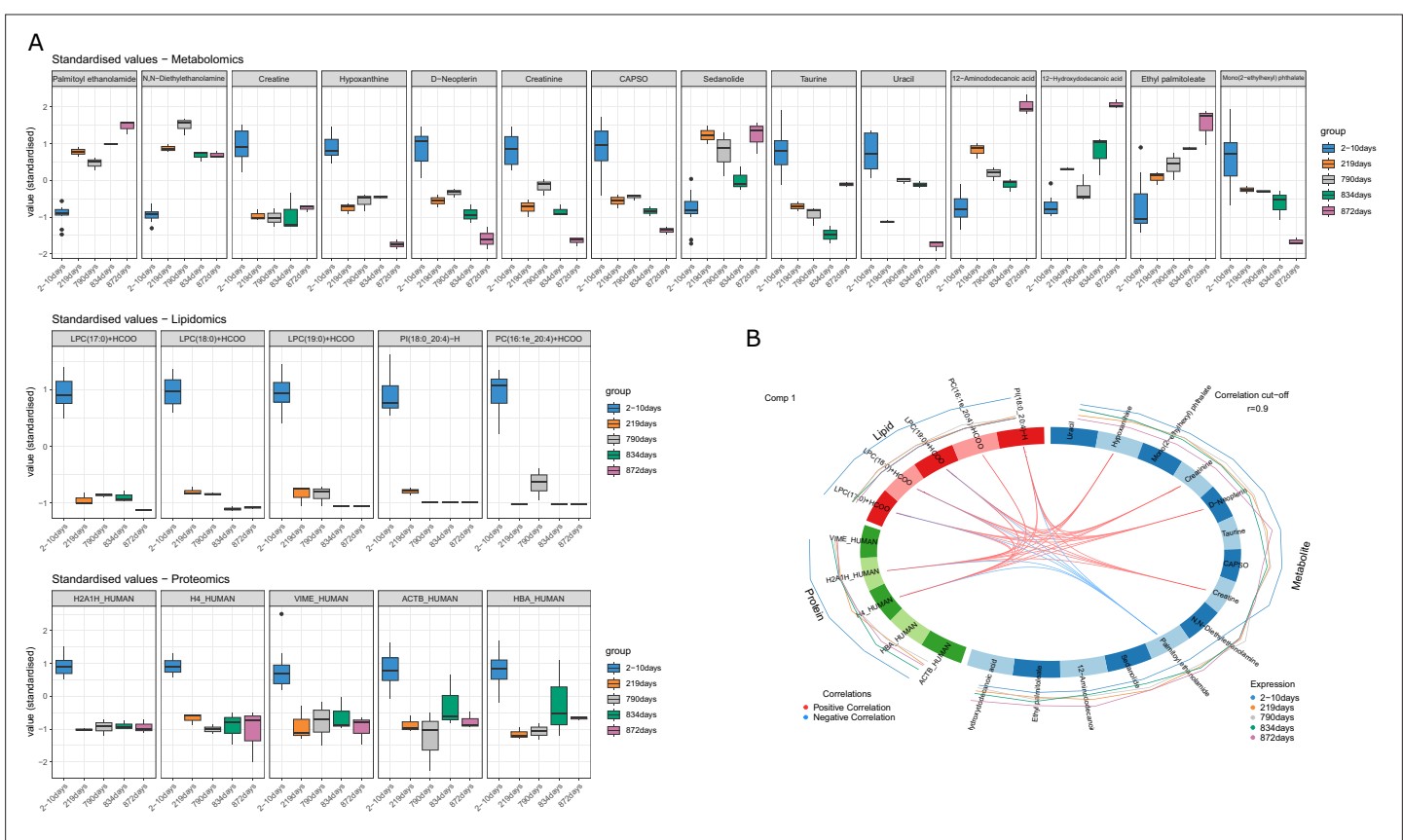

**Figure 2.** DIABLO selected variables correlated with PMI. (**A**) Boxplots of the selected variables after tuning that shows variation with postmortem interval (PMI). Variables are expressed in standardized values. (**B**) Correlation between different omics blocks highlighting the correlations between different compounds obtained with the three omics selected in the final discriminant analysis model.

PC(16:1e_20:4)+HCOO and with H2A1H_HUMAN and H4_HUMAN proteins, whereas creatine was positively correlated with all lipids selected but PC(16:1e_20:4)+HCOO and with H2A1H_HUMAN.

## Discussion

This study comprises, to the best of our knowledge, the first attempt to apply a panel of three omics methods to human bones from a controlled decomposition experiment, to identify potential biomarkers for biomolecular PMI estimation. To develop and validate multi-omics PMI estimation methods for forensic applications, replication studies in substantial sample sizes of human bones will be necessary. However, the availability of bone samples both before and after decomposition from the same individuals is currently very limited. The work presented here represents a proof-of-concept study on the potential advantages of combining different omics for PMI estimation. The small number of individuals included is consistent with numbers generally used in human decomposition experiments, in which for practical and ethical reasons larger samples, such as used in clinical studies, are very difficult to obtain. While the sample size used here is not suitable for validation purposes, it serves to demonstrate the value and potential of the 'ForensOMICS' approach.

Considering each omics individually, the proteomic profile appears to show quite a considerable overlap between the individuals from three post-decomposition groups (i.e., 219, 790, and 834 days) suggesting that this method on its own does not provide sufficient sensitivity to segregate close PMIs (*Figure 1—figure supplement 3*). This could be due to the nature of these biomolecules; proteins, in fact, are highly stable and may be better suitable for long-term PMI estimation in forensic scenarios (*Procopio et al., 2018b*; *Prieto-Bonete et al., 2019*) as well as in the investigation of archaeological remains (*Wadsworth et al., 2017*; *Warinner et al., 2022*). Additionally, other analyses such as post-translational protein modifications may reveal a greater potential for PMI estimation in bones than the evaluation of the abundance of specific markers on their own (*Procopio et al., 2018b*). Employing a system biology approach for PMI estimation for forensic purposes by combining more than one class of biomolecules that have different postmortem stability (*Dent et al., 2004*) provides a more comprehensive biological explanation of the processes under investigation. This is achieved here by combining different layers of omics (i.e., metabolomics, lipidomics, and proteomics) to reconstruct the molecular profile of the overall system. The DIABLO model simultaneously identifies important markers to optimize the classification of a specific variable by combining multiple omics techniques (*Singh et al., 2019*). This is normally used to explain the biological mechanisms that determine a disease and its development, while in our case the main advantage is represented by the potential of selecting a pool of compounds that effectively explains, and could accurately estimate, PMI changes over an extended period of time. One interesting aspect of this approach is the difference in clustering between the metabolite and lipid blocks individually compared to the integration model. It can be seen in *Figure 1—figure supplement 1* (metabolomics block) that samples with increasing PMIs seem to cluster further away from the pre-deposition sample in a time-dependent manner, with the 219 days PMI being closer to the fresh donors and the 872 days one being the furthest located. However, as suggested, the metabolomics profile of D2 seems to be significantly different from the other donors in the fresh state, and this could suggest that interindividual variation could affect the efficient clustering. This has been already highlighted in the proteomics work conducted on the same samples and was likely caused by the health condition of the donor prior to death (*Mickleburgh et al., 2021*). In contrast, the positioning of the PMI in the cluster tree behaves in the opposite way for lipids, where the various profiles seem not to be affected by any apparent interindividual variation in the fresh nor in the decomposed state (*Figure 1—figure supplement 2*). Considering now the clustering of the integrative model, it provides a clear classification of the PMIs obtained by the combination of the three single blocks. Since the approach chosen for this pilot study was discriminant analysis and PMI was provided to the model as a categorical variable, we believe that treating the response variable (PMI) as an ordinal or continuous variable on a larger sample size could improve the interpretation of the results and the forensic applicability of the methodology. Despite acknowledging these limitations, these preliminary results show the possibility of using multi-omics integration to identify different PMI groups. Furthermore, the results for proteomics, that individually does not allow discrimination for these specific time intervals, are integrated in the final model by retaining only the proteins that contribute to PMI identification.

Additionally, the presence of the two main clusters identified (fresh and skeletonized) has been driven by the greater differences between pre- and post-deposition. Conventionally, when performing method development for PMI estimation on bone samples collections, the baseline time is not available. Therefore, the differences captured with the analysis would be obtained on skeletonized samples only. We believe, however, that due to the uniqueness of the sample it was not ideal to remove the pre-deposition specimens. Despite these issues, we found moderate to high correlation between the omics blocks that allows their integration using the sparse algorithm (*Singh et al., 2019*) for PMI estimation.

Recently, literature has grown on the use of molecular studies via omics platforms, especially for short-term PMIs. Most of the studies involving metabolomics for PMI estimation focused on quickly degradable matrices (e.g., muscle, blood, humour) collected over a short period of time (<1 month) (*Pesko et al., 2020*; *Locci et al., 2019*; *Banaschak et al., 2005*; *Ith et al., 2011*; *Ith et al., 2002*). As previously mentioned, the analysis of proteins in bone has shown applicability to estimate relatively long PMIs in forensics (*Procopio et al., 2018b*; *Mizukami et al., 2020*; *Procopio et al., 2018a*) as well as to address archaeological questions (*Ntasi et al., 2022*; *Pal Chowdhury et al., 2021*; *Brandt and Mannering, 2021*; *Richter et al., 2022*; *Brown et al., 2016*), due to the prolonged survival of this type of biomolecules. Finally, according to the studies presented so far, it seems that postmortem changes of lipids could provide PMI estimation across several years, although there is great need for validation (*Dudzik et al., 2017*; *Dudzik et al., 2020*). The combination of these biomolecules' classes in a multi-omics model could therefore be beneficial for estimating PMI across a broader range of potential PMIs. Metabolites and lipids offer accuracy in the short to medium term while proteins could be the main markers for longer PMIs due to their greater stability. Furthermore, variable selection (*Singh et al., 2019*; *Rohart et al., 2017*) would offer the advantage of simplifying experimental procedures and targets those markers that behave consistently with PMI. To limit the potential effects of interindividual variability, we considered variables that showed no outliers among the four body donors and created a model that limits as much as possible the number of predictors without affecting the assessment of the PMI.

Our results for the metabolomics assay display clear differences between the pre- and post-placement bone metabolomic profiles, suggesting the potential to use these profiles to assess long PMIs. The small sample size in this study does not allow us to make any deep inferences about the biological significance of the metabolomics profiles of the post-placement samples, as these may have been influenced by exogenous factors. With regard to the pre-placement samples, the PMIs ranging between 2 and 10 days at 4°C would have allowed some minimal postmortem modifications in the metabolome to occur (*Chighine et al., 2021*). The metabolomic profiles of these samples are characterized by creatine, taurine, hypoxanthine, 3-hydroxybutyrate, creatinine, and phenylaniline. Hypoxanthine is a well-known hallmark of ATP consumption and, consequently, a sign of exhaustion of normal substrates (i.e., glucose and pyruvate) of the tri-carboxylic acid (TCA) cycle. In conjunction with the presence of creatine, taurine, creatinine, phenylalanine, and 3-hydroxybutyrate, we may hypothesize a switch towards TCA cycle anaplerosis through amino acidic and ketonic substrates, in pursuit of a resilient ATP production during the early/mid-PMIs. Not only was the proposed metabolomic approach able to identify the pre- and post-deposition groups according to the bone metabolome modifications, but it was also sensitive enough to detect at very long PMIs. The presence of exogenous compounds (i.e., caffeine, ecgonine, dextromethorphan, tramadol *N*-oxide, penbutolol, salicylic acid) that could reflect lifestyle habits or pharmacological therapies, and thus potentially has major implications in forensic toxicology and personal identification, is consistent with evidence from animal models (*Alldritt et al., 2019*). Enrichment analysis can be found in *Figure 3*.

Several polar metabolites identified in this study have previously been found in other tissues to show a consistent decay pattern after death. In fact, most of the compounds of interest matched here have already been flagged in other tissues as good potential biomarkers of PMI across shorter timeframes (*Figure 2A*). Uracil, a pyrimidine base of RNA, was previously seen to increase over a 14-day PMI in human muscle tissue when analysed by LC-MS (*Pesko et al., 2020*). Similar results for this compound were found in GC-MS analysis of rat's blood (*Dai et al., 2019*). In contrast, no clear association between this metabolite and PMI was found in aqueous humour (*Locci et al., 2019*). In the present study, after a drop in normalized intestines between the baseline and first PMI, we detected an increase until 834 days, and a drop towards the longest PMI considered. It is worth

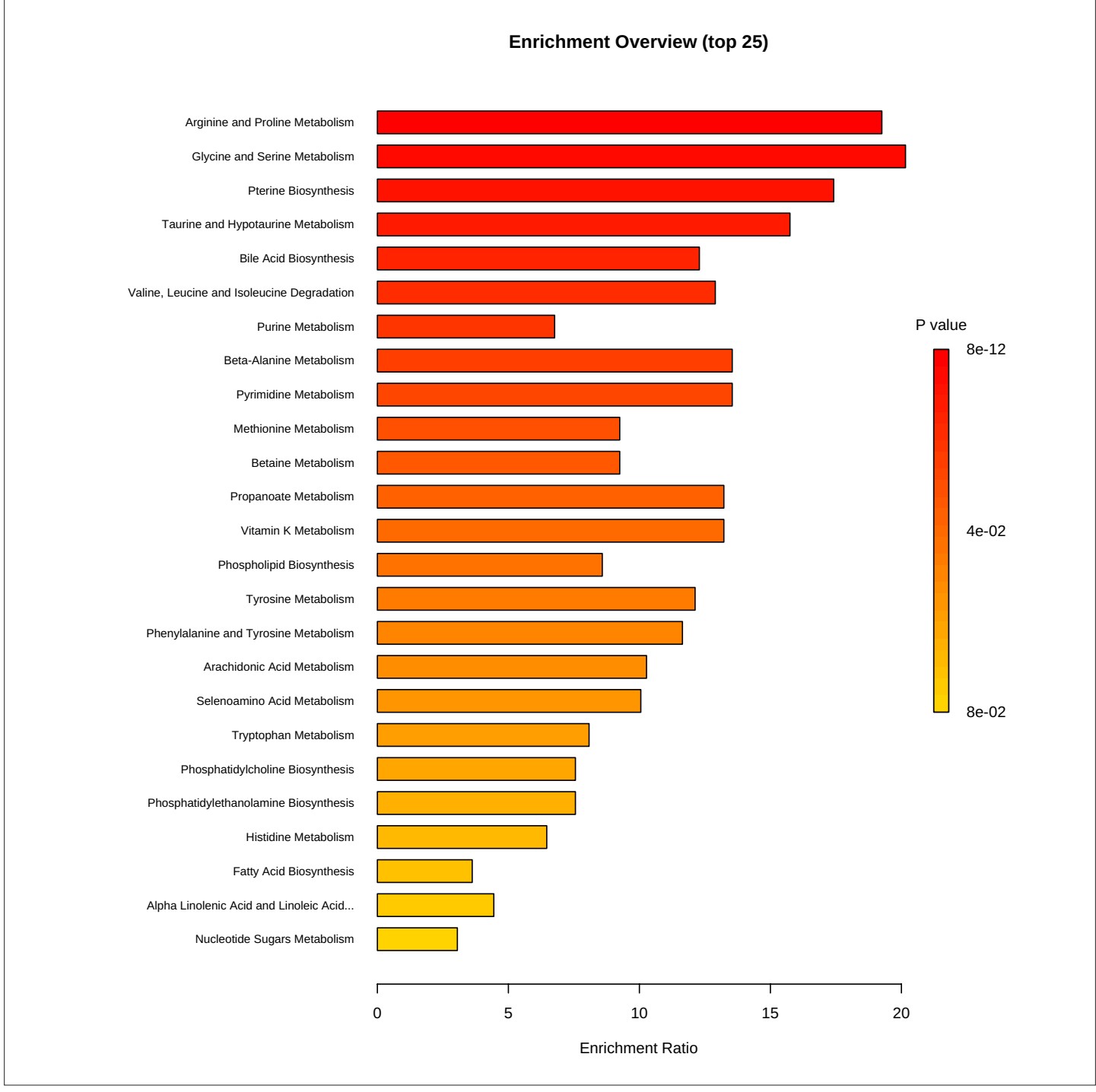

**Figure 3.** Metabolite set enrichment analysis based on differentially expressed metabolites identified in bone.

mentioning that most metabolites drop significantly after the baseline ('fresh') times (*Figure 2A*), suggesting that compound decomposition is driving this first part of the PMI following the stop of human metabolism. It is interesting that with the increase in PMI there is also an increment in several compounds that could be associated with the breakdown of larger biomolecules (e.g., proteins) or with the presence of microbial communities that leave their own metabolic profile on bone surface. Another common marker of interest is hypoxanthine for its association with hypoxia (*Locci et al., 2019*; *Zelentsova et al., 2020*; *Kaszynski et al., 2016*; *Jawor et al., 2019*; *Locci et al., 2021*), which

seems to drastically drop between the baseline times and the first PMI timepoint, as well as in the last time interval, showing a good consistency with PMI. In contrast, hypoxanthine was seen to increase until 48 hr and then to decrease at 72 hr in rat blood (*Donaldson and Lamont, 2013*). *Zelentsova et al., 2020*, showed a positive relation between hypoxanthine and PMI in human serum, aqueous and vitreous humour. To fully understand the behaviour of this compound in bone tissue, a longitudinal study should be performed, also including short PMIs. Leucine has also been reported in short time scale to increase in human muscle tissue (*Pesko et al., 2020*) and this agrees with our results where, after the initial drop, we noticed a consistent increase from the first PMI onwards. What can be clearly seen in *Figure 2A* is that D2 affects the linearity of the trend, suggesting that there might be some degree of interindividual variability. This is the case for several compounds; this limitation could be mitigated by increasing the number of individuals per timepoint in future studies. Creatinine has previously been reported to be a good marker in both muscle tissue (*Pesko et al., 2020*). Although it has not been mentioned in literature previously, we also found that neopterin, a biomarker for immune system activation commonly profiled in blood, serum, and urine (*Melichar et al., 2017*; *Laudanski et al., 2021*), has a strong negative correlation with PMI. Taurine, also in accordance with studies on vitreous humour (*Locci et al., 2019*), showed a predictable positive behaviour with PMI. Acetamide is a nitrogen-based compound associated with active and advanced decay (*Dekeirsschieter et al., 2012*) that, not surprisingly, showed the best positive association with PMI, resulting in being the most reliable biomarker within the entire panel considered. Palmitoylethanolamide is a carboximidic acid that was shown to accumulate in relation with cellular stress in pig brains postmortem (*Buczynski and Parsons, 2010*). These findings agree with our study, which revealed a clear increase of this metabolite with increasing PMIs. *N,N*-diethylethanolamine, belonging to the class of organic compounds known as 1,2-aminoalcohols, has not yet been highlighted for its potential in PMI estimation. In the current study, this molecule increased in the decomposed samples, although no clear trends were observed across the various PMIs. A proposed mechanism for its accumulation is the partial oxidation driven by bacterial decomposition of monosaccharides into organic alcohols (*Dent et al., 2004*; *Nolan et al., 2020*). 12-Aminododecanoic acid and 12-hydroxydodecanoic acid are instead medium-chain fatty acids that show a positive relationship with PMI. Previous studies based on skeletal muscle tissue reported a decline in very-long-chain fatty acids (*Langley et al., 2019*; *Wood and Shirley, 2013*) in very short PMIs. It is not possible to exclude the cleavage of longer chains by the action of lipases or microorganic activity (*Dent et al., 2004*; *Stuart, 2013*). The last compound selected in the final model is methylmalonic acid, a carboxylic acid which is an intermediate in the metabolism of fat and proteins. It has been shown that abnormally high levels of organic acids in blood (organic acidaemia), urine (organic aciduria), brain, and other tissues lead to general metabolic acidosis (*Narayanan et al., 2011*). In this study, even with a postmortem increase in its concentration, it is not possible to identify a clear trend across the decomposed samples; this may be related to interindividual biological differences of the donors involved in this study (e.g., age and health condition).

From the lipidomic assay, only five markers were selected in the final model. These are three lysophosphatidylcholines (LPCs), one phosphatidylcholine (PC) and one phosphatidylinositol (PI), all showing decreasing intensities in the decomposed samples in comparison with the 'fresh' ones. PCs are generally the most abundant neutral phospholipids and represent the main constituent in cellular membranes. LPCs are derived from the hydrolysis of dietary and biliary PCs and are absorbed as such in the intestines, but they become re-esterified before being exported in the lymph (*McMaster, 2018*). They are present in cell membranes and in blood. Their half-life in vivo is limited because of the quick metabolic reaction that involves lysophospholipases and LPC-acyltransferases (*Law et al., 2019*). In contrast, PLS are amphiphilic molecules that are also minorly present in cell membranes, whose role is to modulate the membrane curvature and to have other bioactive functions such as interacting with peripheral proteins (*Falkenburger et al., 2010*) and inhibiting osteoclast formation (*Alhouayek et al., 2018*). After death, these compounds can be converted into fatty acids via hydrolysis to then hydrogenize or oxidase to form saturated and unsaturated fatty acids (*Dent et al., 2004*). This process is driven by intrinsic tissues lipases (*Dent et al., 2004*). A very limited number of studies have applied lipidomics for PMI estimation. *Langley et al., 2019*, evaluated human skeletal muscle tissue from 31 donors over a PMI of 2000 accumulated degree days showing consistent extraction of phosphatidylglycerol (PG) 34:0 and phosphatidylethanolamine 36:4, which showed good correlation with PMI. *Wood and Shirley, 2013*, investigated the lipidome of human anterior quadriceps muscle from one

donor at 1-, 9-, and 24-day PMIs showing the decline of sterol sulphates, choline plasmalogens, etha-nolamine plasmalogens, and PGs and the increase of free fatty acids. Our results lend support to these earlier findings and further confirm the potential of lipidomics for PMI estimation. Nonetheless, direct comparison with these studies is not possible as they considered different tissues for much shorter PMIs. Additionally, lipids profiled from the muscle tissue after decomposition are suggested to derive from cell membrane breakdown (*Langley et al., 2019*; *Wood and Shirley, 2013*). We suggest that, in bone material, the lipidome under investigation accounts not only for cell membrane decomposition of embedded osteocytes but also for the marrow and fluids embedded in the bone pores.

The proteomics results revealed that two ubiquitous proteins (histones), haemoglobin, ACTB, and VIME are the best candidates within this multi-omics PMI model. These five proteins selected by the model represent those which were best able to discriminate between the 'fresh' bones and the 'skeletonized' bones but are therefore not necessarily the best biomarkers to differentiate between the four post-decomposition PMIs. For insights on the most suitable protein biomarkers for differentiating between the longer PMIs, identified by excluding the 'fresh' samples, see *Mickleburgh et al., 2021*. It is not surprising to see that the proteins highlighted in the model are either ubiquitous proteins or blood or muscle tissue proteins, as their abundance would naturally be higher in 'fresh' bone than in 'skeletonized' bones. The HBA is found in red blood cells but is often also identified in bone samples with long PMIs from archaeological contexts (*Smith and Wilson, 1990*), and its consistent time-dependent degradation has been previously highlighted in skeletal remains using several platforms (*Ramsthaler et al., 2011*; *Wiley et al., 2009*). Furthermore, it has already been reported in skeletal tissue from controlled decomposition studies of animals, and already highlighted as a potential biomarker for PMI estimation (*Procopio et al., 2018b*). VIME was also previously reported by *Procopio et al., 2018b*, to be associated with PMI. It is a filament protein abundant in muscle tissue, and therefore its association with bone, particularly with the 'fresh' samples, is not unexpected. However, we emphasize that this could also be due to interindividual variability, and that further investigation may clarify the usefulness of VIME to estimate PMI. ACTB, similar to VIME, is a structural protein that forms cross-linked networks in the cytoplasmatic compartments and that is strongly connected with the presence of muscle tissue residues. A previous study showed the decrease in myosin contents with increasing PMIs, similarly to what we observed here for ACTB. The remaining two proteins are both components of the nucleosomes, in our study were shown to be drastically reduced in bone tissue also at the first the baseline PMI taken into consideration. In sum, these results allowed the identification of five protein biomarkers which make good candidates for estimation of short PMIs (<900 days) (e.g., considering time points limited to months postmortem) and not for years after death for which structural and functional proteins in bone have been shown better targets to employ for PMI estimation (*Mickleburgh et al., 2021*; *Prieto-Bonete et al., 2019*).

Based on the findings of this exploratory study, we argue that the multi-omic method we adopted here shows considerable potential for the future development of an accurate and precise PMI estimation method for human bone. Further research should focus on increasing the sample size, to ultimately validate the method for application in forensic investigation of skeletonized human remains. Beyond the findings discussed at length above, we emphasize that it is of paramount importance to establish which biomolecules identified here are associated with the human metabolism and degradation, and which are produced by the decomposers' microbial activity. Controlled taphonomic experiments on human decomposition at human taphonomy facilities provide the opportunity to elucidate biomolecular decomposition of human bone. A comprehensive understanding of the origin of different compounds is key to provide a detailed explanation of the postmortem changes that affect bone and other tissues, ultimately helping to shed a light on biomolecular PMI investigations and on the real potential that multi-omics analyses can have in this direction.

## Materials and methods
### Body donors
Bone samples were collected from four female human body donors, aged between 61 and 91 years (mean 74±11.6 SD), at the Forensic Anthropology Center at Texas State University (FACTS). FACTS receives whole body donations for scientific research under the Texas revised Uniform Anatomical Gift Act (*Health and Safety Code, 2009*). Body donations are made directly to FACTS and are exclusively

**Table 1.** Sample composition, demographics, deposition context, and postmortem interval (PMI). The sample ID column reports the biological replicates used. Additional information on the body donors and observations made during collection of bone samples (e.g., medical treatments, bone colour, and density) can be found in the supplementary information in *Mickleburgh et al., 2021*.

| Sample ID | Sex | Age (years) | PMI | Deposition context |
|---|---|---|---|---|
| Pre-deposition samples | | | | |
| D1_TF_A | Female | 91 | 10 days | Open pit |
| D1_TF_B | Female | 91 | 10 days | Open pit |
| D1_TF_C | Female | 91 | 10 days | Open pit |
| D2_TF_A | Female | 67 | 2 days | Burial |
| D2_TF_B | Female | 67 | 2 days | Burial |
| D2_TF_C | Female | 67 | 2 days | Burial |
| D3_TF_A | Female | 61 | 3 days | Burial |
| D3_TF_B | Female | 61 | 3 days | Burial |
| D3_TF_C | Female | 61 | 3 days | Burial |
| D4_TF_A | Female | 77 | 10 days | Open pit |
| D4_TF_B | Female | 77 | 10 days | Open pit |
| D4_TF_C | Female | 77 | 10 days | Open pit |
| Post-deposition samples | | | | |
| D1_TS_A | Female | 91 | 219 days | Open pit |
| D1_TS_B | Female | 91 | 219 days | Open pit |
| D1_TS_C | Female | 91 | 219 days | Open pit |
| D2_TS_A | Female | 67 | 834 days | Burial |
| D2_TS_B | Female | 67 | 834 days | Burial |
| D2_TS_C | Female | 67 | 834 days | Burial |
| D3_TS_A | Female | 61 | 790 days | Burial |
| D3_TS_B | Female | 61 | 790 days | Burial |
| D3_TS_C | Female | 61 | 790 days | Burial |
| D4_TS_A | Female | 77 | 872 days | Open pit |
| D4_TS_B | Female | 77 | 872 days | Open pit |
| D4_TS_C | Female | 77 | 872 days | Open pit |

acquired through the expressed and documented will of the donors and/or their legal next of kin. Demographic, health, and other information are obtained through a questionnaire completed by the donor or next of kin. The data are securely curated by FACTS, and the body donation program complies with all legal and ethical standards associated with the use of human remains for scientific research in the United States. The number of individuals (n=4) used in this preliminary study is consistent with other taphonomic studies conducted on human remains for proof-of-concept purposes. Larger sample sizes may be used to validate preliminary results, such as those proposed by this study, at a later stage.

The bodies were stored in a cooler at 4°C prior to sampling. After collection of the initial (pre-placement) bone samples, the bodies were placed outdoors to decompose at the Forensic Anthropology Research Facility (FARF), the human taphonomy facility managed by FACTS, between April 2015 and March 2018. Two of the four body donors (D1 and D4, see *Table 1*), were placed in shallow hand-dug pits which were left open throughout the duration of the decomposition experiment. The

pits were covered with metal cages to prevent disturbance by large scavengers. Donors D2 and D3 were deposited in similarly sized hand-dug pits and were immediately buried with soil. Environmental data for the duration of the project are available as *Supplementary file 2*.

## Sampling

Bone samples (ca. 1 cm³) of the anterior midshaft tibia were collected prior to placement of the body outdoors, and again upon retrieval of the completely skeletonized remains as can be seen in *Figure 4*. Each body was in 'fresh' stage of decomposition when pre-placement samples were taken, and in 'skeletonization' stage when post-placement samples were collected, based on scoring of the gross morphological changes (*Megyesi et al., 2005*). The duration of each placement and the deposition context are reported in *Table 1*. The soft tissue was incised with a disposable scalpel, and a 12 V Dremel cordless lithium-ion drill with a diamond wheel drill bit was used at max. 5000 revolutions to collect ~1 cm³ of bone. Sampling instruments were cleaned with bleach and deionized water between each individual sample collection.

A total of eight samples were collected in Ziploc bags, transferred immediately to a –80°C freezer, and subsequently shipped overnight on dry ice to the Forensic Science Unit at Northumbria University, UK. The samples were then transferred to a lockable freezer at –20°C as per UK Human Tissue Act regulations (licence number 12495). Part of the analyses were conducted by the 'ForensOMICS' team (NP and AB) at Northumbria University prior to their transfer to the University of Central Lancashire. Specifically, the bone samples were defrosted, and fine powder was obtained with a Dremel drill equipped with diamond-tipped drill bits operated at speed 5000 rpm, to avoid heat damage caused by the friction with the bone. The collected powder was homogenized and stored in 2 mL protein LoBind tubes (Eppendorf UK Limited, Stevenage, UK) at –80°C until extraction and testing. The powder sample was later divided into 25 mg aliquots. Three biological replicates (e.g., three aliquots of bone sample per specimen) were extracted and analysed for each specimen. The research and bone sample analyses were reviewed and approved by the Ethics committee at Northumbria University (ref. 11623).

## Biphasic extraction, adapted Folch protocol

Chloroform (Chl), AnalaR NORMAPUR ACS was purchased from VWR Chemicals (Lutterworth, UK). Water Optima LC/MS Grade, Methanol (MeOH) Optima LC/MS Grade, Pierce Acetonitrile (ACN), LC-MS Grade and Isopropanol (IPA), OptimaLC/MS Grade were purchased from Thermo Scientific (Hemel Hempstead, UK). In total three biological replicates for each of the eight specimens were extracted according to a modified (*Folch et al., 1957*) as follows: 25 mg of bone powder was placed in tube A and 750 µL of 2:1 (v/v) Chl:MeOH were added, vortexed for 30 s, and sonicated in ice for additional 20 min. Three-hundred µL of LC-MS grade water was added to induce phase separation and sonicate for another 15 min. The sample was then centrifuged at 10°C for 5 min at 2000 rpm. The lower (lipid) fractions were collected and transferred to fresh Eppendorf tubes and the samples were re-extracted with a second time using 750 µL of 2:1 (v/v) Chl:MeOH. The two respective fractions were combined and the remaining aqueous fractions centrifuged at 10°C for 5 min at 10000 rpm and the supernatant tranferred to fresh Eppendorf tubes. The organic lipid fraction was preconcentrated using a vacuum concentrator at 55°C for 2.5 hr or until all organic solvents have been removed. The aqueous metabolite fractions were flash frozen in liquid nitrogen and preconcentrated using a lyophilizer cold trap –65°C to remove all water content. The respective dry fractions were then stored at –80°C until analysis. The metabolite fraction was resuspended in 100 µL in 95:5 ACN/water (% v/v) and sonicated for 15 min and centrifuged for 15 min at 15 K rpm at 4°C and supernatant was then transferred to 1.5 mL autosampler vials with 200 µL microinsert and caped. Twenty µL of each sample were collected and pooled to create the pooled QC. The lipid extracts were resuspended in 100 µL of 1:1:2 (v/v) water:ACN:IPA and sonicated for sonicated for 15 min and centrifuged for 15 min at 15 K rpm at 10°C and supernatant was then transferred to 1.5 mL autosampler vials with 200 µL microinsert and caped. Twenty µL of each sample were collected and pooled to create the pooled QC. The sample set was then submitted for analysis.

## LC-MS analysis

Metabolite and lipid characterization of the bone samples was performed on a Thermo Fisher Scientific (Hemel Hempstead, UK) Vanquish Liquid Chromatography (LC) Front end connected to IDX High

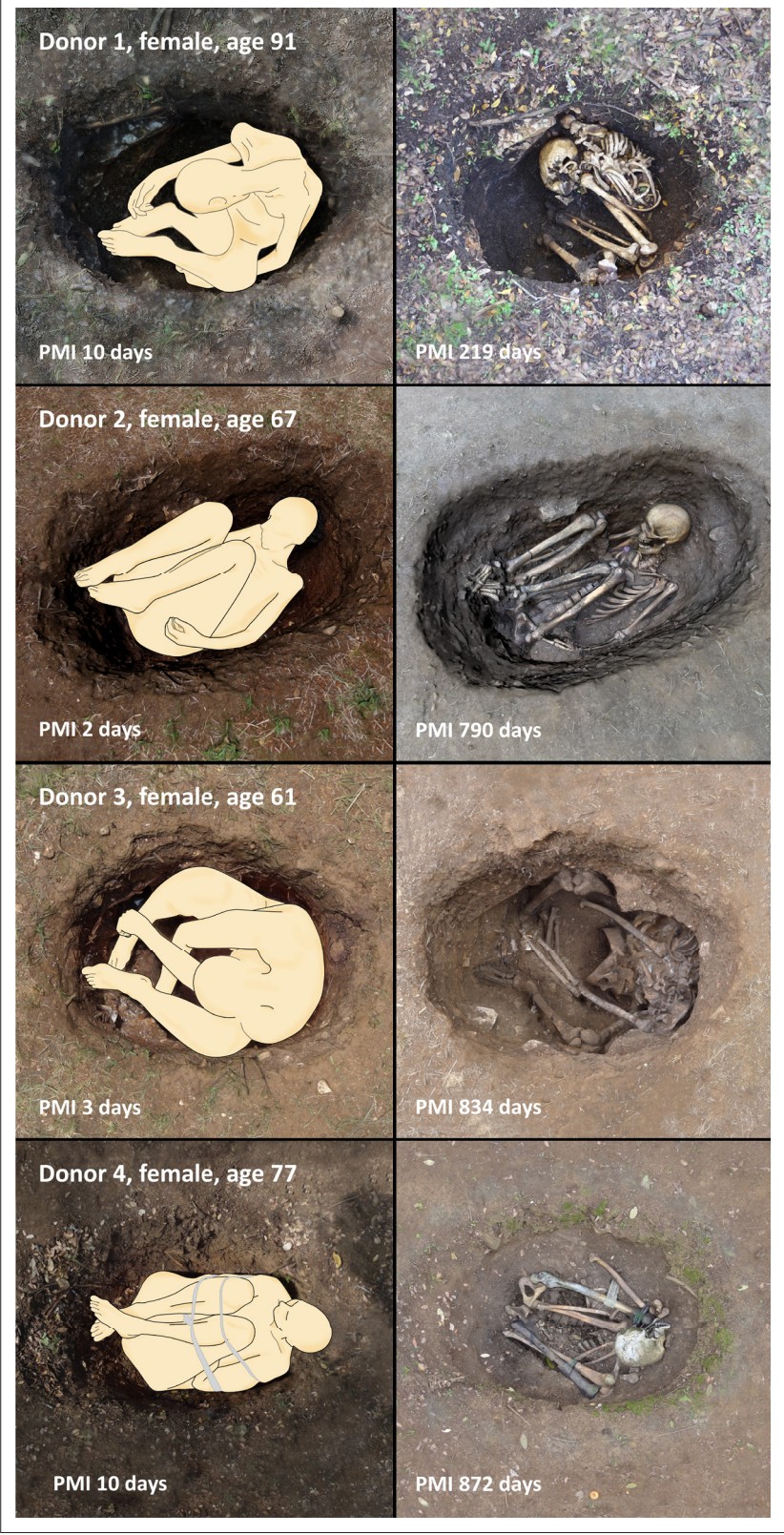

**Figure 4.** Positioning of the bodies in the single graves (left) pre-decomposition and (right) after complete skeletonization.

The online version of this article includes the following figure supplement(s) for figure 4:

**Figure supplement 1.** Flowchart of the experimental design of the study.

Resolution Mass Spectrometer (MS) system. Full details for both metabolomics and lipidomics runs are reported below.

## Metabolomics

Hydrophilic liquid interaction chromatography was used for the chromatographic separation for metabolites. The separation was achieved using a Waters Acquity UPLC BEH amide column (2.1×150 mm with particle size of 1.7 μm, part no. 186004802), operating at 45°C with a flow rate of 200 μL/min. The LC gradient consists of a binary buffer system, namely buffer 'A' (LC/MS grade water) and buffer 'B' (LC/MS grade ACN) both containing 10 mM ammonium formate. Independent buffer systems were used for positive and negative electrospray ionization (ESI) acquisition respectively, for ESI+ the pH of buffers was adjusted using 0.1% formic acid and for negative using 0.1% ammonia solution. The LC gradient was the same for both polarities, namely 95% 'B' at T0 hold for 1.5 min and a linear decrease to 50% 'B' at 11 min, followed by hold for 4 min, return to starting condition and hold for further 4.5 min (column stabilization). The voltage applied for ESI+ and ESI- was 3.5 and 2.5 kV, respectively. Injection volumes used were 5 μL for ESI+ and 10 μL for ESI-.

## Lipidomics

Standard reverse phase chromatography was used for the chromatographic separation of lipids. The separation was achieved using a Waters Acquity UPLC CSH C18 column (2.1×150 mm with particle size of 1.7 μm, part no. 186005298), operating at 55°C with a flow rate of 200 μL/min. The LC gradient consists of a binary buffer system, namely buffer 'A' (LC/MS grade water:ACN, 40:60 % v/v) and buffer 'B' (IPA:ACN, 90:10% v/v) both containing 10 mM ammonium formate. Independent buffers systems were used for positive and negative ESI modes respectively, for ESI+ the pH of buffers was adjusted using 0.1% formic acid and for negative using 0.1% ammonia solution. The LC gradient was the same for both polarities, namely 60% 'B' at T0 hold for 1.5 min, linear increase to 85% 'B' at 7 min, increase to 95% 'B' at 12.5 min and hold for 4.5 min before returning to starting conditions and holding for further 4.5 min (column stabilization). The voltage applied for ESI+ and ESI- was 3.5 and 2.5kV, respectively. Injection volumes used were 3 μL for ESI+ and 5 μL for ESI-.

The HESI conditions for 200 μL were as follows: sheath gas 35, auxiliary gas 7, and sweep gas of 0. Ion transfer tube temperature was set at 300°C and vaporizer temperature at 275°C. These HESI conditions were applied to both metabolomics and lipidomics and lipidomics assays.

## MS acquisition

MS data were acquired using the AcquieX acquisition workflow (data-dependent analysis). The MS operating parameters were as follows: MS1 mass resolution 60 K, for MS2 30 K, stepped energy (HCD) 20, 25, 50, scan range 100–1000, RF len (%) 35, AGC gain, intensity threshold $2^4$, 25% custom injection mode with an injection time of 54 ms. An extraction blank was used to create a background exclusion list and a pooled QC was used to create the inclusion list.

## Data processing

The metabolomic positive and negative data sets were processed via Compound Discoverer (version 3.2) using the untargeted metabolomic workflow with precursor mass tolerance 10 ppm, maximum shift 0.3 min, alignment model adaptive curve, minimum intensity $1^6$, S/N threshold 3, compound consolidation, mass tolerance 10 ppm, RT tolerance 0.3 min. Database matching were performed at MS2 level using Thermo Scientific mzCloud mass spectral database with a similarity index of 50% or higher.

The lipidomic positive and negative data sets were processed via Thermo Scientific LipidSearch (version 4) using the following workflow: HCD (high energy collision database), retention time 0.1 min, parent ion mass tolerance 5 ppm, product ion mass tolerance 10 ppm. Alignment method (max), top rank off, minimum m-score 5.0, all isomer peaks, ID quality filter A and B only. Lipid IDs were matched using LipidSearch in silico library at MS2 level. Corresponding metabolomics and lipidomics pooled QCs samples were used to assess for instrumental drifts; the relative standard deviation (RSD) variation across the QCs for metabolomics and lipidomics were less than 15%. Any metabolite/lipid feature with an RSD of 25% or less within the QCs was retained.

## Proteomics

Proteomics results from a pilot study conducted on the same samples used in this study were previously published and discussed in *Mickleburgh et al., 2021*. Analyses were conducted following an adapted protocol developed by *Procopio and Buckley, 2017a*, for protein extraction and LC-MS-MS analysis. MS data for proteomic analysis were made available via ProteomeXchange Consortium via the PRIDE (*Ternent et al., 2014*) partner repository with the data set identifier PXD019693 and 10.6019/PXD019693.

## Statistical analysis

An overview of the ForensOMICS pipeline can be found in *Figure 4—figure supplement 1*. Metabolomics and lipidomics data were normalized by mean values, cube root transformation and Pareto scaling was applied. Proteomics data were normalized using log2 transformation. For preliminary data evaluation, Principal component analysis (PCA) was applied to the profiles obtained by each single chromatographic separation method for metabolomics and lipidomics and for the proteomic block to exclude data sets with poor discriminatory power. At first, univariate analysis was performed by Kruskal-Wallis. Despite the small sample size per PMI, pairwise Dunn's test with Holm's corrected p-value was applied to the set to have an overview of the differences between different PMIs. PLS-DA was first employed to analyse each omics block. Correlation between blocks was then investigated with pairwise PLS regression prior to DIABLO analysis (*Singh et al., 2019*) based on multiblock sPLS-DA using the 'mixOmics' package in R (version 4.1.2) (*Rohart et al., 2017*). The initial model was tuned using a threefold/100 repeats cross-validation to perform variable selection and produce a final model that maintains the maximum covariance reducing the number of the compounds used for the classification. Classification error rate was further cross-validated (threefold, 100 repeats) and significance of the classification was tested via permutation test (k=3 and 999 permutation) implemented in the 'RVAideMemoire' package (*Hervé et al., 2018*). All cross-validation in this study was performed considering explicitly the biological replicates. Enrichment analysis was carried out considering pre- and post-placement samples combined.

## Conclusions

In conclusion, our results support the potential for developing an accurate and precise multi-omics PMI estimation method for human bone for application in forensic contexts to aid criminal investigation and assist with identification of the deceased. Despite the small sample size used here, this study demonstrates how the approach can discriminate between short and long PMIs. This method can produce classification models including different markers (e.g., protein, metabolites, and lipids) to assess both short- and long-term PMIs, with a high level of accuracy, as the compounds under investigation have complementary decay rates. The use of different biochemical markers that have different postmortem stability offers the advantage of covering both short-term PMIs, by including metabolites and lipids, and long-term PMIs, by implementing in the model more stable proteins that consistently degrade after death. This could not be fully proven based on our results, as the PMI taken into exam is not sufficiently spread along the timeline and more individuals per timepoint are necessary. However, the possibility of selecting only discriminating variables allows the combination of omics that in isolation could not discriminate in a satisfactory way the PMI. In the present study, proteomics did represent the less ideal omics for the estimation of the time elapsed since death, however few protein variables were successfully included in the model. Furthermore, in the present study the order between the various PMIs was voluntarily not considered in data analysis in order to avoid biases in the generation of the discriminant model. We expect that the PMI estimation over extended time periods will be unlikely achieved by employing any of these three omics individually. Furthermore, treating PMI as a continuous variable could be key in providing an optimal approach for the estimation of PMI. Furthermore, this methodology provides new insights on the biological processes that occur after death and will help establishing whether the presence of certain molecules is the result of their molecular degradation or if it is mostly associated with the bacterial metabolism, a central question in forensic science. The proposed 'ForensOMICS' approach must be validated by the analysis of substantial sample sizes in future controlled taphonomic experiments conducted in multiple different environments, as this represents the main source of variation in human decomposition, as well as by

evaluating a broader PMI with a more comprehensive coverage of data points in the time period taken into consideration.

## Acknowledgements

The authors acknowledge the UKRI for supporting this work by the UKRI Future Leaders Fellowship (NP) under grant MR/S032878/1, as well as the European Research Council (grant 319209) and the Leiden University Fund (grant 5604/30-4-2015/Byvanck) for supporting the actualistic taphonomic experiment at FARF. We would also like to thank the NUOmics Facility at Northumbria for the MS analyses and data pre-processing, and the donors and their next of kin for allowing the use of donated bodies to perform this research.

## Additional information

### Funding

| Funder | Grant reference number | Author |
|---|---|---|
| UK Research and Innovation | MR/S032878/1 | Noemi Procopio |
| European Research Council | 319209 | Hayley L Mickleburgh |
| Leiden University | 5604/30-4-2015/Byvanck | Hayley L Mickleburgh |

The funders had no role in study design, data collection and interpretation, or the decision to submit the work for publication.

### Author contributions

Andrea Bonicelli, Conceptualization, Data curation, Software, Formal analysis, Validation, Investigation, Visualization, Methodology, Writing – original draft, Writing – review and editing; Hayley L Mickleburgh, Conceptualization, Supervision, Funding acquisition, Writing – original draft, Project administration, Writing – review and editing; Alberto Chighine, Emanuela Locci, Data curation, Writing – original draft, Writing – review and editing; Daniel J Wescott, Conceptualization, Writing – original draft, Writing – review and editing; Noemi Procopio, Conceptualization, Resources, Formal analysis, Supervision, Funding acquisition, Validation, Methodology, Writing – original draft, Project administration, Writing – review and editing

### Author ORCIDs

Andrea Bonicelli http://orcid.org/0000-0002-9518-584X
Hayley L Mickleburgh http://orcid.org/0000-0001-9326-8097
Alberto Chighine http://orcid.org/0000-0003-0952-9712
Emanuela Locci http://orcid.org/0000-0003-0237-4228
Noemi Procopio http://orcid.org/0000-0002-7461-7586

### Ethics

This study did not involve human living subjects but only human tissues from deceased individuals who gave their consent prior to death. The institutional review board is not required in these circumstances. The ethics code of FACTS was adhered and an approval for the study was obtained from the FACTS Ethics review board as well as from the Ethics committee at Northumbria University (ref. 11623) and at UCLan (ref. SCIENCE 0223).

### Decision letter and Author response

Decision letter https://doi.org/10.7554/eLife.83658.sa1
Author response https://doi.org/10.7554/eLife.83658.sa2

## Additional files

### Supplementary files
- Supplementary file 1. Univariate analyses for all the individual omics.
- Supplementary file 2. Environmental data.
- MDAR checklist
- Source code 1. R pipeline employed in the study.

### Data availability
This data is available at the NIH Common Fund's National Metabolomics Data Repository (NMDR) website, the Metabolomics Workbench, with Study ID ST002283. The data can be accessed via Project DOI https://doi.org/10.21228/M8MH6X. The R script used is available in *Source code 1*.

The following dataset was generated:

| Author(s) | Year | Dataset title | Dataset URL | Database and Identifier |
|---|---|---|---|---|
| Bonicelli A | 2022 | ForensOMICS | https://doi.org/10.21228/M8MH6X | Metabolomics Workbench, 10.21228/M8MH6X |

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
