## [Editor Report]

This well-presented and sophisticated study provides significant proof-of-concept for the application of the ForensOMICS approach as a new pathway for forensic taphonomy with great promise to advance future research. The solid foundation of the research combining metabolomics, proteomics, and lipidomics is considered very exciting, strong, and expands the boundaries of forensics research.

---

## [Decision Letter]

**Decision letter after peer review:**

Thank you for submitting your article "The "ForensOMICS" approach to forensic post-mortem interval estimation: combining metabolomics, lipidomics and proteomics for the analysis of human bone." for consideration by *eLife*. Your article has been reviewed by 3 peer reviewers, and the evaluation has been overseen by a Reviewing Editor and Mone Zaidi as the Senior Editor. The following individual involved in the review of your submission has agreed to reveal their identity: Sharni Collins (Reviewer #1).

Essential revisions:

Please address the following concerns/points:

1. Please consider the inclusion of placement periods/seasons of each of the donors.

Do you collect any environmental/covariate data, like daily temperature, relative humidity, rainfall, etc of the taphonomic field site? Going forward, with future studies, this information could be extremely useful to inform any future conclusions. In addition, it could open the door to linear mixed models where you could investigate the impact of random or fixed effects, alongside the outcomes of the omics results. You could also potentially use this approach to investigate inter-individual variability, as mentioned in your work. This may be more of a long-term goal, but it is definitely something worth looking into.

2. pg. 5, line 127: "so the metabolomic appears…" suggestion → "so the metabolomic approach appears…".

3. pg. 6, line 173: "PMI estimation methods is has not…" suggestion → "PMI estimation methods has not…".

4. From a methodological point of view, as a first step of data analysis, each single data set (metabolomics, lipidomics, and proteomics) should be investigated singularly, independently of the other two data sets, to discover the information of each single data set related to PMI and better understand the advantage to integrate the three data sets. The analysis should be performed by multivariate and univariate data analysis. Moreover, the common and unique information of the three data sets should be investigated in order to better understand what happens when the data are integrated.

5. PMI is a continuous factor, not a categorical factor. I suppose that clustering analysis has been performed in the latent space obtained by modelling the data considering all three data sets and that the discovered groups of similar observations have been characterized by PLS-DA. How many clusters have been obtained? What classes have been used in PLS-DA? If this was the approach, the order between PMI values have not been considered in data analysis. PMI was modeled as a categorical factor instead of a continuous factor or, eventually, an ordinal factor. Moreover, it seems that only two main groups of observations have been discovered and that these two groups correspond to pre- and post-burial. This may be the effect of a different time scale between the three -omics. For example, if metabolomics and lipidomics explain low PMI but not high PMI and proteomics explain only high PMI, only two clusters can be observed by data integration. The analysis suggested at point 1 may clarify the situation. Please, comment on these points and add a brief description of DIABLO in Supplementary Information.

6. Cross-validation must be performed considering explicitly the biological replicates. All replicates of the same subject at the same time must be excluded during each run of cross-validation, i.e. they must belong to the same group. Please repeat cross-validation if this was not the case.

7. The model should be tested by randomization test to discover over-fitting. Please, discuss this point or implement a suitable permutation test.

8. There are garbled codes in the R script file provided by the authors, which seem to be line breaks. Please check the encoding of the script.

9. It is suggested to use a flow chart to describe the general process of the "ForensOMICS" pipeline, instead of only visually displaying the results of different omics analyses, so that readers can intuitively and clearly understand the main content of the study.

10. Significance analysis could be performed in figure 3 to test the significant variations of the selected variables between different PMI.

11. There are still some mistakes in the writing, further revisions and corrections are needed.

12. Consistent with the well-received transparency of the authors' approach pertaining to different aspects of this study, It would also be beneficial to the readership to comment on weaknesses raised by reviewer #3.

*Reviewer #1 (Recommendations for the authors):*

Overall, a very sophisticated paper that has been both written and presented well. It is exciting to see what the future holds as the field moves towards more robust scientific applications, such as those presented in this work. It is great to see the transparency in the work as well, understanding that much work is still yet to be done before this approach is operational. However, this does not detract from the significance of the work at all. What I think would be beneficial to include in the current paper, if you choose to accept, is the inclusion of placement periods/seasons of each of the donors. This is common practice for some laboratories, and can sometimes be really useful in informing my next point.

Do you collect any environmental/covariate data, like daily temperature, relative humidity, rainfall, etc of the taphonomic field site? Going forward, with future studies, this information could be extremely useful to inform any future conclusions. In addition, it could open the door to linear mixed models where you could investigate the impact of random or fixed effects, alongside the outcomes of the omics results. You could also potentially use this approach to investigate interindividual variability, as mentioned in your work. This may be more of a long-term goal, but it is definitely something worth looking into.

*Reviewer #2 (Recommendations for the authors):*

The authors should address the following main points.

1. From a methodological point of view, as a first step of data analysis, each single data set (metabolomics, lipidomics, and proteomics) should be investigated singularly, independently of the other two data sets, to discover the information of each single data set related to PMI and better understand the advantage to integrate the three data sets. The analysis should be performed by multivariate and univariate data analysis. Moreover, the common and unique information of the three data sets should be investigated in order to better understand what happens when the data are integrated.

2. PMI is a continuous factor, not a categorical factor. I suppose that clustering analysis has been performed in the latent space obtained by modelling the data considering all three data sets and that the discovered groups of similar observations have been characterised by PLS-DA. How many clusters have been obtained? What classes have been used in PLS-DA? If this was the approach, the order between PMI values have not been considered in data analysis. PMI was modelled as a categorical factor instead of a continuous factor or, eventually, an ordinal factor. Moreover, it seems that only two main groups of observations have been discovered and that these two groups correspond to pre- and post-burial. This may be the effect of a different time scale between the three -omics. For example, if metabolomics and lipidomics explain low PMI but not high PMI and proteomics explain only high PMI, only two clusters can be observed by data integration. The analysis suggested at point 1 may clarify the situation. Please, comment on these points and add a brief description of DIABLO in Supplementary Information.

3. Cross-validation must be performed considering explicitly the biological replicates. All replicates of the same subject at the same time must be excluded during each run of cross-validation, i.e. they must belong to the same group. Please repeat cross-validation if this was not the case.

4. The model should be tested by randomization test to discover over-fitting. Please, discuss this point or implement a suitable permutation test.

*Reviewer #3 (Recommendations for the authors):*

1. There are garbled codes in the R script file provided by the authors, which seem to be line breaks. Please check the encoding of the script.

2. It is suggested to use a flow chart to describe the general process of the "ForensOMICS" pipeline, instead of only visually displaying the results of different omics analyses, so that readers can intuitively and clearly understand the main content of the study.

3. Significance analysis could be performed in figure 3 to test the significant variations of the selected variables between different PMI.

4. There are still some mistakes in the writing, further revisions and corrections are needed.

---

## [Author Response]

Reviewer #1 (Recommendations for the authors):Overall, a very sophisticated paper that has been both written and presented well. It is exciting to see what the future holds as the field moves towards more robust scientific applications, such as those presented in this work. It is great to see the transparency in the work as well, understanding that much work is still yet to be done before this approach is operational. However, this does not detract from the significance of the work at all. What I think would be beneficial to include in the current paper, if you choose to accept, is the inclusion of placement periods/seasons of each of the donors. This is common practice for some laboratories, and can sometimes be really useful in informing my next point.

We thank the reviewer for their opinion on the robustness and novelty of our multi-omics work. We agree that this is a useful information and therefore we included it in the main text of the resubmitted manuscript.

Do you collect any environmental/covariate data, like daily temperature, relative humidity, rainfall, etc of the taphonomic field site? Going forward, with future studies, this information could be extremely useful to inform any future conclusions. In addition, it could open the door to linear mixed models where you could investigate the impact of random or fixed effects, alongside the outcomes of the omics results. You could also potentially use this approach to investigate interindividual variability, as mentioned in your work. This may be more of a long-term goal, but it is definitely something worth looking into.

We agree with the reviewer. We have therefore provided environmental conditions gathered during the experiment in the Supplementary File 2. Unfortunately, it was not possible to include the weather parameters in the statistical analysis (eg linear mixed model) as the number of samples is limited, but this will be taken into consideration for our future studies on a larger sample size.

Reviewer #2 (Recommendations for the authors):The authors should address the following main points.1. From a methodological point of view, as a first step of data analysis, each single data set (metabolomics, lipidomics, and proteomics) should be investigated singularly, independently of the other two data sets, to discover the information of each single data set related to PMI and better understand the advantage to integrate the three data sets. The analysis should be performed by multivariate and univariate data analysis. Moreover, the common and unique information of the three data sets should be investigated in order to better understand what happens when the data are integrated.

We agree that analysing each block separately could give a better understanding of their contribution to the final model. Univariate statistics was therefore added for each single omics (based on Kruskal-Wallis and Dunn Pairwise analysis with Holm corrected p-value). Multivariate statistic for each block was added by using PLSDA. Limitations of pairwise univariate analysis were highlighted in the text. We have added three plots and a table in supplementary material to discuss the results for all the individual omics. Univariate analysis for the proteomics is largely discussed in a previous publication (10.1021/acs.jproteome.6b01070). However, due to this not being the main purpose of the manuscript, we only briefly presented these results mostly focusing in comparing them with the integrative model.

Concerning the performance of each single omics block, we found that metabolomics, at least for this PMI range, is probably a better target for future estimations on relatively short PMIs. However, here we mainly aimed to evaluate the potential of combining blocks and we speculate that, on wider PMIs, the combination of more than one omics could be the best approach, granted the validation of the model. New figures can be found in Figure 1 —figure supplements 1-4.

2. PMI is a continuous factor, not a categorical factor. I suppose that clustering analysis has been performed in the latent space obtained by modelling the data considering all three data sets and that the discovered groups of similar observations have been characterised by PLS-DA. How many clusters have been obtained? What classes have been used in PLS-DA? If this was the approach, the order between PMI values have not been considered in data analysis. PMI was modelled as a categorical factor instead of a continuous factor or, eventually, an ordinal factor. Moreover, it seems that only two main groups of observations have been discovered and that these two groups correspond to pre- and post-burial. This may be the effect of a different time scale between the three -omics. For example, if metabolomics and lipidomics explain low PMI but not high PMI and proteomics explain only high PMI, only two clusters can be observed by data integration. The analysis suggested at point 1 may clarify the situation. Please, comment on these points and add a brief description of DIABLO in Supplementary Information.

We agree that the approach could not be ideal for the estimation of a continuous phenomenon such as PMI. Unfortunately, the small sample size and the distribution across different PMIs led us to avoid regression approaches that will be the way to go in future studies. We believe, and specified in the text, that the clustering is mostly driven by the two macro groups (fresh vs. decomposed) and therefore this leads to the loss of some information on the decomposition process. However, the rarity of the sample including pre and post decomposition specimens is such that the fresh ones could not be excluded from the analysis. We believe that, adding the single omics results, has been useful to better address this concern. The rendered and explained version of the DIABLO pipeline and script has been uploaded as a separate supplementary source code file 1. A detailed description of DIABLO is available in the cited work (doi: 10.1093/bioinformatics/bty1054) referenced in the main text.

3. Cross-validation must be performed considering explicitly the biological replicates. All replicates of the same subject at the same time must be excluded during each run of cross-validation, i.e. they must belong to the same group. Please repeat cross-validation if this was not the case.

We thank the reviewer for this very useful comment. The replicates have now been excluded during each run of cross-validation and the results have been corrected accordingly. This can be verified in the script. We also decide to use set.seed() for reproducibility.

4. The model should be tested by randomization test to discover over-fitting. Please, discuss this point or implement a suitable permutation test.

A permutation test based on repeatedCV (k=3, perm=999) was implemented and the weighted predicted classification error rate was reported in the text. Furthermore, we estimated the classification error rate by cross-validation (repeatedCV k=3, times=100). Data are reported in the Results section.

Reviewer #3 (Recommendations for the authors):1. There are garbled codes in the R script file provided by the authors, which seem to be line breaks. Please check the encoding of the script.

The code has been corrected and rendered for upload.

2. It is suggested to use a flow chart to describe the general process of the "ForensOMICS" pipeline, instead of only visually displaying the results of different omics analyses, so that readers can intuitively and clearly understand the main content of the study.

We have added the flow chart as per request and added it to Figure 4 —figure supplement 1.

3. Significance analysis could be performed in figure 3 to test the significant variations of the selected variables between different PMI.

We agree with the reviewer that significance values old represent an addition to the manuscript, but we are aware that the small sample size does not provide sufficient robustness. Despite this, we included a section containing univariate statistic (Kruskal-Wallis and Dunn pairwise test with Holms p-value correction) that can be accessed in the Supplementary File 1 to avoid overcrowding the plots.

4. There are still some mistakes in the writing, further revisions and corrections are needed.

We have revised the test and we have corrected all mistakes, to the best of our knowledge.